# Prevention of the Onset of Age-Related Macular Degeneration

**DOI:** 10.3390/jcm10153297

**Published:** 2021-07-26

**Authors:** Emiliano Di Carlo, Albert J. Augustin

**Affiliations:** Department of Ophthalmology, Städtisches Klinikum Karlsruhe, 76133 Karlsruhe, Germany; albertjaugustin@googlemail.com

**Keywords:** age-related macular degeneration, prevention, nutrients, lifestyle

## Abstract

Age-related macular degeneration (AMD) represents the leading cause of irreversible blindness in elderly people, mostly after the age of 65. The progressive deterioration of visual function in patients affected by AMD has a significant impact on quality of life and has also high social costs. The current therapeutic options are only partially able to slow down the natural course of the disease, without being capable of stopping its progression. Therefore, better understanding of the possibilities to prevent the onset of the disease is needed. In this regard, a central role is played by the identification of risk factors, which might participate to the development of the disease. Among these, the most researched are dietary risk factors, lifestyle, and light exposure. Many studies showed that a higher dietary intake of nutrients, such as lutein, zeaxanthin, beta carotene, omega-3 fatty acids and zinc, reduced the risk of early AMD. Regarding lifestyle habits, the association between smoking and AMD is currently accepted. Finally, retinal damage caused by ultraviolet rays and blue light is also worthy of attention. The scope of this review is to summarize the present knowledge focusing on the measures to adopt in order to prevent the onset of AMD.

## 1. Introduction

Age-related macular degeneration (AMD) represents nowadays the leading cause of irreversible blindness in elderly people, mostly after the age of 65 [1]. Recent studies predict that by 2040, the number of affected individuals worldwide will be circa 288 million [2]. In addition, the progressive increase in life expectancy of the world’s population and, in particular, that of developing countries, will lead to a further increase in the elderly population [3]. By 2025, it has been estimated that the geriatric population will reach the amount of circa 1.2 billion people worldwide, 70% of whom residing in developing countries [4]. This process may contribute strongly to a steady increase in the number of people suffering from AMD, even greater than that assumed by previous studies.

AMD is a retinal disease that typically affects the macula, causing progressive loss of central vision [5]. Early-stage AMD is mainly characterized by clinical signs such as drusen and alterations of the retinal pigment epithelium. Late-stage AMD can assume two different forms: neovascular, also known as wet or exudative, or non-neovascular, also named as atrophic, dry, or non-exudative. AMD progression to late stages results in loss of central vision, leading to severe and persistent visual impairment and legal blindness, thus resulting in a significant impact on the quality of life and also hindering the functional independence of affected individuals [6].

Early AMD is clinically characterized by typical disturbs, such as mild central distortion and reduced reading capacity with decreased luminance. Difficulty to clearly recognize faces and central scotoma are also additional signs that can appear. Nevertheless, in this stage, affected people are mostly asymptomatic. Late AMD may progress to a sudden and rapid deterioration in case of a neovascular form, while a slow and gradual decline of central visual function is typical of the atrophic form. The main feature of neovascular AMD (nAMD) is the presence of choroidal neovascularization and its repercussions, such as intra- and subretinal fluid, hemorrhages, pigment epithelial detachments, hard exudates and, finally, fibrotic tissue. Geographic atrophy is characterized by outer retinal thinning, and it has been evaluated that the progression rate corresponds to circa 2 mm^2^/year on average [7]. It is important to point out that, according to the previous reports and classifications, what will be described regarding the prevention of AMD is relative to the “drusen-driven” AMD, without considering the spectrum of pachychoroid diseases.

Nowadays, although the use of intravitreal anti-vascular endothelial growth factor VEGFinjections has contributed considerably to improve the treatment of nAMD, no effective treatment has been yet found for both early AMD and geographic atrophy. The absence of a therapy that might at least stop the unavoidable progression of the clinical course determines a functional limitation in daily activities and, consequently, a worsening of the quality of life for the patients.

In the last years, many researchers have tried to define modifying risk factors, especially how to address retinal oxidative changes related to both onset and progression of the disease. In particular, positive behavioral modifications including smoking cessation, exercise, and healthy diet with the addition of nutritional supplements [8,9].

The attention of these studies has been mainly focused on the ability to intervene on the progression of early AMD towards more advanced stages of the disease and geographic atrophy, thereby modifying the aforementioned risk factors. 

The aim of this review is to focus the attention on all the modifiable risk factors that might help to prevent, slow, or reduce the onset of AMD in healthy individuals.

## 2. Prevalence and Incidence of AMD

The most relevant information regarding the epidemiology of AMD has been provided by three large, population-based studies: the Blue Mountains Eye Study (BMES), Beaver Dam Eye Study (BDES), and Rotterdam Study (RS), in which data were collected and analyzed, mainly related to both incidence and prevalence of AMD in white populations [10].

The above-mentioned studies have found a prevalence of late AMD of 0.2% for people aged between 55 and 64 years, while an increased prevalence (13.1%) has been observed for people over 85 years [11]. Specifically, BMES demonstrated for early AMD a 15-year incidence of 22.7% and an incidence of 6.8% for late AMD, with an overall incidence greater in women rather than men [12]. At the same time, a meta-analysis from the European Eye Epidemiology Consortium including 14 population-based cohort studies evidenced a prevalence of 13.2% for early AMD and 3.0% for late AMD for people with more than 70 years [13].

Another large meta-analysis reported a higher prevalence of early AMD in European white people when compared with Asian population (11.2% and 6.8%). Moreover, in comparison to African people, there is a major prevalence of both early and late AMD for European population, while no differences have been found between African and Asian populations. On the contrary, the prevalence of nAMD has been estimated to be similar in all ethnic groups (0.46%) [2].

Regarding the prevalence of AMD among different populations, it is noteworthy to underline that both population-based and the Age-Related Disease Eye Study AREDS studies were performed before the concept of pachychoroid spectrum diseases had been widely recognized. Specifically, the difference between pachychoroid neovasculopathy and “drusen-driven” neovascular AMD has not been exhaustively analyzed. Pachychoroid spectrum diseases are much common in Asian compared to Western populations. Asians currently constitute 60% of the world’s population and likely will contribute most greatly to the global prevalence of AMD by 2040. Before the introduction of the concept of pachychoroid spectrum diseases, pachychoroid neovasculopathy had often been diagnosed as AMD especially in Asian populations. This may partly explain why in the previous population-based studies, there was no difference in the prevalence of neovascular AMD in different ethnic groups even though early AMD is more common in Western than in Asian populations [14].

The different prevalence of non-exudative AMD among different ethnic groups may be partly explained by genetic differences. With regard to this, a genetic study has demonstrated a lower frequency of the C-risk allele of the Y402H polymorphisms in Japanese as compared with white populations [15].

## 3. AMD Impact on Quality of Life and Social Costs

AMD may reduce quality of life with a significant impact on the performance of normal daily activities. The burden of AMD in terms of health has been analyzed with the aim of the disability-adjusted life years (DALY), which represents an indicator that measures the loss of years of life due to AMD [16,17].

Reports exhibited increased levels of life stress and depression in patients affected by AMD compared to healthy subjects, mostly when results of treatment do not correspond to patients’ expectations [18,19]. In addition, it has been demonstrated in patients affected by AMD a greater risk of functional disability, as showed by BMES, where a two times higher risk of negative effects on daily living activity was observed [20].

An interesting paper has evaluated the correlation between AMD and the risk to develop a form of cognitive impairment, including Alzheimer’s disease, throughout the course of life. The researchers found an increased risk for AMD patients to suffer from cognitive impairment and Alzheimer’s disease, mostly in case of atrophic AMD [21].

Finally, it has been estimated by a large meta-analysis that AMD is associated with a 20% increased risk of overall mortality and, specifically, a 46% increased risk for cardiovascular disease [22].

A deepened understanding of global patterns in health burden related to AMD is fundamental to implement focused strategies of prevention. In this regard, a recent study has estimated the social costs of blindness related to AMD in 2020 in the United States. Specifically, the authors analyzed excess costs that occur because of blindness, measuring the differences in total costs between blind and non-blind individuals. They found that the annual amount of excess costs for each blind individual is about USD 5000. Translated to the whole society, it means a total societal cost of circa USD 20 billion, a value that is estimating to triple by 2050 [23].

In the last 25 years, the health burden of AMD has continuously grown without pause worldwide. Despite the introduction of anti-VEGF therapy that has radically improved the clinical history of at least exudative AMD, the DALY due to AMD has not accordingly improved [16]. This process has been exhaustively analyzed by previous reports across countries, which estimated that the proportion of individuals affected by both visual impairment and blindness due to AMD raised by 81% and 36% between the years 1990 and 2010, respectively [24]. These observations demonstrate that the socio-economic burden related to AMD has not been relieved with the introduction of an effective therapy. For this reason, due to the challenge determined by the increase of both prevalence and burden of AMD, it may be necessary for the next years to arrange additional resources on a global scale to fight this phenomenon.

## 4. AREDS Studies: Interventions to Stop AMD Progression

A key role in the pathophysiology of AMD progression is played by oxidative stress. Cellular aging is mainly characterized by the production of oxygen free radicals, which eventually leads through various mechanisms to cell death. The retina has a proper defensive mechanism against oxidative processes, which consists of vitamins C and E, carotenoids, lutein, and zeaxanthin [8]. In addition, the presence of a major structural lipid, docosahexaenoic acid (DHA), at the level of cones, is involved in membrane permeability and acts against the formation of new vessels [25]. Based on these abovementioned findings, researchers have made attempts over the years to develop a therapy with nutritional supplements that can hinder an unstoppable process, such as AMD.

As early as the late 1990s, attempts have been made to evaluate the impact of adding nutrients to diet that could somehow slow or stop the progression of the disease toward blindness [26]. In 1988, Newsome et al. [27] demonstrated that a diet with zinc supplementation might reduce the visual acuity loss in patients affected by AMD. In addition, other researchers assessed the efficacy of vitamin formulations containing zinc and antioxidants for AMD, finding contrasting results [28].

The most important reports relative to the nutritional supplementation therapy have been shown by the two AREDS studies [29,30]. The AREDS study assessed the effect of high-dose vitamins C and E (500 mg and 400 IU, respectively), β-carotene (15 mg), and zinc supplementation (80 mg) on AMD progression. The patients included in the study were divided into four groups based on disease’s gravity, ranging from early to advanced AMD. The study demonstrated that AREDS supplementation reduced the risk of AMD progression as compared to placebo. However, there was no significant effect on the risk of visual acuity loss. It was also recommended that nutrition supplements should only be used by people who do not smoke, because subsequent analyses of other studies has found a greater incidence of cancer in smokers or recent ex-smokers who assumed β-carotene [31].

The AREDS-2 clinical trial [30] was conducted with the aim to ameliorate the efficacy of the AREDS formulation. For this reason, a new formulation with the addition of lutein and zeaxanthin, which has the maximum concentration at the fovea, plus DHA and EPA (docosahexaenoic and eicosapentaenoic acid) was evaluated in order to find a therapy for decreasing the risk of AMD progression. Moreover, the new study also assessed the effect of the new formulation after eliminating β-carotene and adjusting the dose of zinc to only 25 mg. Therefore, it has been proved that the risk of AMD progression was not more reduced as compared to the previous AREDS study, and, in addition, it was suggested that the first formulation should have included lutein and zeaxanthin, without β-carotene supplementation. Nowadays, the recommended formulation based on AREDS-2 results for dry AMD contains vitamin E (400 IU), vitamin C (500 mg), lutein (10 mg), zeaxanthin (2 mg), copper (2 mg), and zinc (80 mg, but also available with the 25 mg formulation) (Table 1).

## 5. Prevention of AMD Onset: Dietary Modification and Nutritional Supplementation; Smoking and Lifestyle Modifications; Role of Blue-Light, UV Radiation, and Intraocular Lenses

### 5.1. Dietary Modification and Nutritional Supplementation 

Both AREDS-1 and -2 studies have shown to prevent AMD progression by nutritional supplementation. However, there are no data regarding the possibility to prevent the onset of AMD, either by modifying dietary habits or by adding nutritional supplements to the diet.

The Rotterdam study of 2005 [32] evaluated whether the intake of antioxidants with the diet was associated with a reduced risk to develop AMD. A questionnaire was used to estimate the dietary intake for a total of 5836 people at risk of AMD, but without signs of pathology in either eye. Specifically, at baseline, participants had to complete a checklist at home that queried foods and drinks they had consumed at least twice a month, dietary habits, use of nutritional supplements, and prescribed diets. Finally, these data were assembled and transformed to total energy intake per day with the computerized Dutch Food Composition Table [33]. The study found that a high consumption with the diet of β-carotene, both vitamins C and E, and zinc was correlated with a decreased risk of AMD in elderly persons.

The formation of drusen is mainly influenced by oxidative protein modifications, as shown by previous studies [34]. Since the onset of drusen is the first clear visible sign of AMD, it is logical to argue that the action of antioxidant molecules exerts its maximum effectiveness just before the onset of the disease. Therefore, as suggested by the Rotterdam study, a diet rich in antioxidants might hinder the onset of AMD and it should be recommended mostly for those people with a strong family history [35]. Nutrient sources of vitamin E are whole grains, vegetable oil, eggs, and nuts. A higher intake of zinc can be achieved by consuming meat, poultry, fish, whole grains, and dairy products. β-carotene can be found in carrots, kale, and spinach. Citrus fruits and juices, green peppers, broccoli, and potatoes are the main suppliers of vitamin C (Table 2).

Recently, the EYE-RISK Consortium [36] investigated the relationship between the adherence to the Mediterranean diet (MeDi) and the incidence of advanced AMD in two European population-based prospective cohorts: the Rotterdam Study 1 (RS-1) [37] and the Antioxydants, Lipides Essentiels, Nutrition et Maladies Oculaires (Alienor) [38]. Specifically, compliance to MeDi was assessed with the aim of a specific a score based on consumption of vegetables, fruits, legumes, cereals, fish, meat, dairy products, and alcohol, and the monounsaturated-to-saturated fatty acids ratio. The authors concluded that a regular adherence to the MeDi was correlated with a 41% reduction of the risk to develop advanced AMD. These findings support the role of a diet rich in healthful nutrient-rich foods such as fruits, vegetables, legumes, and fish for preventing AMD onset. In addition, the study has suggested that also reducing unhealthful foods, such as red and processed meats and savory and salty industrialized products, might give a strong contribution to prevent the onset of AMD.

The rationale behind the choice to analyze a specific type of diet, such as the Mediterranean one, can be explained by the fact that a single nutrient or food approach is not able to reach synergistic effects of food and nutrients consumed in combination in the diet. High intake of plant foods and fish, lower intake of meat and dairy products, olive oil as the main source of fat, and a moderate consumption of wine represents the cornerstones of MeDi [39] (Table 3). The results of the EYE-RISK Consortium are also in accordance with those of previous cross-sectional studies. For example, the Carotenoids in Age-Related Eye Disease Study (CAREDS) study has demonstrated a decreased prevalence of early AMD in American women who have strongly adhered to the MeDi [40]. In addition, reports from the Coimbra study have shown a reduced prevalence of any type of AMD in participants who properly adhered to the MeDi [41]. Lastly, the European Eye Study has proven that high levels of MeDi score are associated with a lower prevalence of nAMD [42].

The role of lutein and zeaxanthin has been exhaustively studied in the last years not only for their properties to slow down the progression of AMD, but also to assess their capability for preventing AMD onset [43]. Delcourt et al. [44] conducted a prospective study on a total of 640 individuals with age over 60 who were affected by different forms of AMD to ascertain whether there exists an association between dietary lutein and zeaxanthin intake and the manifestation of the disease. The authors found a strong inverse correlation between serum levels of lutein and zeaxanthin and the presence of AMD, suggesting a protective role played by the two xanthophylls. These results have been later confirmed by the study of Huang et al. [45], which revealed that participants with high levels of serum lutein and zeaxanthin had a 79% reduced risk to develop AMD in comparison to subjects with lower serum concentration. Furthermore, the prevention role carried out by the xanthophylls was assessed in The Blue Mountains Eye Study [46], where it has been demonstrated that persons who assume higher amounts of lutein and zeaxanthin show a reduced risk to develop drusen or AMD. Dietary sources of lutein and zeaxanthin are represented by chicken egg yolk and leafy green vegetables, whose main sources are kale, parsley, spinach, and broccoli (Table 3). Moreover, it is possible to find trace amounts of the two xantophylls in wheat and grain products, like corn, einkorn wheat, and durum wheat.

High food consumption of omega 3 long-chain polyunsaturated fatty acids with the diet, such as docosahexaenoic acid (DHA) and eicosapentaenoic acid (EPA), has been demonstrated to be linked with a reduced risk of AMD onset [47,48]. Omega 3 fatty acids allow the continuous renewal of RPE cells and, if absent, may contribute to degradation of photoreceptor structures and accumulation of drusen at the level of RPE [49]. Specifically, it has been demonstrated that high DHA levels of oily fish are correlated with less risk to develop choroidal new vessels [50]. In addition, the Nutritional AMD Treatment 2 (NAT2) Trial revealed that individuals with high blood levels of both EPA and DHA were significantly protected against AMD as compared to those with lower content of omega 3 [50,51]. However, a Cochrane meta-analysis asserted that there is no evidence to support that nutritional supplementation with omega 3 is able to prevent the onset of AMD [52]. Despite the discordant results that emerged in the scientific literature and the recent Cochrane meta-analysis, it might be reasonable for the physician to inform the patients about the positive effects of consuming fatty fish or omega 3 fatty acid supplements to prevent AMD onset.

Calcium regulation plays an important role in AMD onset, just as for other neurodegenerative diseases, including glaucoma, Alzheimer’s, and Parkinson’s disease [53]. The association between daily calcium intake and AMD onset has been extensively analyzed by Kaigi et al. in a cross-sectional study [54]. The authors found that individuals who consumed daily more than 800 mg of calcium showed a significantly higher probability to have AMD, although a dose relationship was not observed. It is noteworthy that the 800 mg/day cut off is lower than the daily intake of calcium for men and women in USA [55]. For these reasons, it should be necessary to conduct other studies in order to better evaluate the relation between high calcium consumption and AMD onset.

### 5.2. Smoking and Lifestyle Modifications

Cigarette smoking has been associated with the onset of AMD and represents a strong predictor of disease in two population-based longitudinal study [56]. A review by Thornton et al. [57] came to the conclusion that cigarette smoking represents a fundamental risk factor for the AMD onset, estimating with precision that the risk is circa 2- to 3-fold higher in smokers in comparison to nonsmokers. The EUREYE study [9] is a cross-sectional study that analyzed the relationship between cigarette smoking and both type and bilaterality of AMD in individuals aged 65 and older in European countries with the use of a single protocol for eye examinations and the evaluation of risk factors. The study reported an increased risk of neovascular AMD for smokers, and the attributable factor for AMD due to smoking was 27%. Interestingly, when compared with subjects affected by unilateral AMD, those with a bilateral form of AMD were more likely to be heavy smokers in the previous 25 years. Based on these findings, it is noteworthy to underline the possible risks to develop different forms of AMD associated with smoking and the benefit of quitting smoking. In particular, patients unilaterally affected by AMD who are current smokers should be informed about the possible risk to develop the disease in the other eye. A report of the Beaver Dam Eye Study [58] examined the association of pack-years smoked to incidence of AMD, showing that a greater number of pack-years smoked was associated with an increased risk of transforming from no AMD to minimal early AMD. The precise mechanism that determines the onset of AMD due to smoking has not yet been elucidated, but it seems that oxidative stress and alterations of both retinal and choroidal blood flow are involved [59].

Excessive or reduced sleep duration has been related with various negative health conditions, such as total mortality, cardiovascular diseases, diabetes, and arterial hypertension [60]. In this regard, Khurana et al. [61] studied the association between sleep patterns and AMD. A total of 1003 patients were surveyed for past sleep histories with the aim of a specific questionnaire and the patients’ retina graded as no AMD, early AMD, neovascular AMD, and geographic atrophy. The authors concluded that longer sleep duration (more than 8 h) was correlated with a 7-fold higher risk of geographic atrophy. Nevertheless, it is also relevant to underline how it is only an association and not a cause–effect relationship. However, in order to take action to prevent the onset AMD it is important to consider the possibility that excessive sleep duration may contribute to the onset of AMD and, specifically, geographic atrophy. In addition, Perez-Canales et al. [62] investigated the association between self-reported sleep duration and neovascular AMD (nAMD). Four categories of sleep duration were identified: <6 h, 6–7 h, 7–8 h, and >8 h. The study demonstrated a higher prevalence of short sleep duration (<6 h) in subjects affected by nAMD as compared with age- and sex-matched controls. Moreover, the researchers observed a trend towards an increased risk to manifest nAMD in individuals with short and long sleep duration in comparison to those in the 7–8 h group, indicating a possible U-shaped relationship between sleep duration and nAMD.

Recent reports have recognized excess body weight and obesity as fundamental risk factors for cardiovascular disease [63]. Furthermore, overweight may determine different physical changes, such as a higher level of oxidative stress and inflammatory processes and an imbalance of blood lipids, all factors which are implicated in the pathophysiological mechanisms of AMD onset [64]. Previous studies proved that excess body fat can alter the transport and deposition of carotenoids from blood to macula, thus leading to a reduction of macular pigment levels at the fovea [65]. In the AREDS study, an increased risk of geographic atrophy was observed in individuals with higher BMI. Interestingly, Zhang et al. [66] evaluated the association between categories of body mass index (BMI) and AMD risk in different stages. Seven prospective cohort studies with 1613 cases identified among 31,151 subjects have been analyzed. The authors concluded that excess body weight and obesity are associated with a higher risk to develop AMD and, above all, it has been demonstrated that people affected by obesity have a major risk to manifest late AMD in comparison to subjects with a normal weight. Moreover, a potential linear relationship between BMI and AMD risk was found, suggesting that maintaining normal body weight and preventing excessive weight gain might protect against the onset of AMD. Nevertheless, it has been demonstrated that the waist–hip ratio (WHR) and waist circumferences represent more reliable indicators of abdominal obesity compared to BMI. Further to this point, a decrease in WHR was found to be associated with reduced risk of AMD [67].

An increasing number of scientific reports evidence the positive influence of physical activity on morbidity and mortality, a better preservation of cognitive functions, and reducing biomarkers of aging [68]. Regarding the relationship between physical activity and AMD, studies demonstrated a protective role of physical activity towards AMD onset [69,70]. Nevertheless, current evidence is inconclusive because previous studies were conducted in nonhomogeneous populations with different systems of AMD classification and, in addition, the value of physical activity was not assessed uniformly. A recent systematic review and meta-analysis [71] investigated the association of physical activity and AMD among Caucasian population. The authors concluded that physical activity was associated with lower odds of both early and late AMD among Caucasians, suggesting that staying active throughout life might prevent the onset of AMD. The amount of activity considered as an active lifestyle was as little as three hours of moderate- to low-intensity physical activity per week, which reinforces the concept that also a small amount of physical activity may protect against the development AMD. It has been demonstrated that regular exercise might ameliorate both the activity of antioxidant enzymes and resistance to oxidative stress [72]. For these reasons, the increase in oxidative resistance resulting from moderate physical activity might lead to AMD prevention.

In conclusion, a recent review revealed that a healthy diet without food rich in sugar, fat, alcohol, refined starch, and oils, absence of smoking, and moderate physical activity are associated with a reduced risk of AMD [50]. Moreover, interestingly, the risk reduction was found greater when multiple lifestyles were considered together. 

Public health interventions with the aim to adopt a correct lifestyle, including healthy diet, physical activity, and cessation of smoking, should be recommended strategies for AMD prevention (Table 4). These suggestions are crucial for people with genetic risk and also a family history of AMD.

### 5.3. Role of Blue-Light, UV Radiation and Intraocular Lenses

The scientific literature has clearly shown that excessive exposure to light may damage surface tissues, such as the eye and the skin. For this reason the damage determined by sunlight exposure can also be involved in the onset of AMD [73]. In this regard, a fundamental role in the process of tissue damage is played by the so-called “blue light”. The term blue light refers to a high-energy, short-wave visible light of 400–500 nm. It has been demonstrated that blue light is able to induce a photochemical damage of the retina because of the existence of photosensitizers in this specific wave-band [74]. Human healthy adults have developed special defense systems to protect eye structures from blue light damage. Indeed, visible light passes through the crystalline lens and macular pigments, which have the function to absorb blue light. The crystalline lens has the capability to absorb higher amounts of both ultraviolet (UV) radiations and visible short-wave light, becoming yellow and accumulating oxidative damage with age, furnishing an increased protection from blue light [75]. In addition, the macular pigments, such as meso-zeaxanthin and lutein and zeaxanthin from dietary intake, represent natural intraocular filters that are able to absorb wavelengths between 400 and 520 nm, with an absorption peak at 460 nm [76]. High-energy UV radiation and blue light are mainly present in natural sunlight, while sources for UV radiation are represented by welding and UV lamps, and artificial sources for blue light include electronic devices and indoor lights [77].

The sun produces three different types of UV rays: UVA, UVB, and UVC. They have a wavelength, respectively, between 320 and 400 nm, 290 and 320 nm, and with a range of 200–290 nm [78]. UVB rays are those that can more likely cause damage to the retina and promote the onset of AMD. They target retinal pigment epithelium RPE cells with various mechanisms, such as direct DNA damage, oxidative stress, and activation of several pathways, among which are NLRP3, MAPK and JAK/STAT, which lead to inflammation, apoptosis, and cell death [79].

The introduction of blue light-filtering (BLF) intraocular lenses (IOLs) has allowed to have an intraocular lens able to filter short-wave light and to selectively decrease the transmission of UV radiation. It has been observed that IOLs not capable to absorb UV radiation might lead to significant retinal damage and, thus, facilitate AMD onset [80]. The addition of short-wave light filtering differentiates BLF-IOLs from other lenses because the presence of this filter makes them more structurally similar to the normal crystalline lens [81]. The rationale behind the use of BLF IOLs is that replacing the cataractous natural lens with an IOL without blue-light filter causes an unnatural condition, in which a high amount of blue light is transmitted inside the eye, producing the release of A2E, a short-wave photosensitizer that could damage the retina and RPE cells [82].

Regarding the possibility of preventing AMD onset, an interesting study has been published by Nagai et al. [83]. The authors examined the influence of BLF IOLs in the development of AMD, measuring changes in fundus autofluorescence after implantation of BLF (yellow-tinted) and UV-filtering (colorless) IOLs. They observed a significantly higher incidence of AMD in patients who received a UV-filtering IOL (11%) as compared with those where a BLF IOL was implanted (2%) 2 years after the implantation. These results show how it may be possible that the use of BLF IOLs could prevent the onset of AMD.

Despite the strong pathophysiological basis and studies that have demonstrated a favorable role of BLF IOLs in protecting against the onset of AMD, other data contradict these conclusions. For example, a recent analysis of systematic reviews about BLF IOLs for retinal protection has shown that benefits of using BLF IOLs are not currently supported by the best available research evidence, suggesting that surgeons keep in mind this limitation when adopting these devices in clinical practice [84]. In addition, recently, Achiron et al. [85] has assessed the effect of BLF IOLs on the prevention of nAMD after cataract surgery. In a large cohort study, including 11,397 eyes of 11,397 patients who received BLF and non-BLF IOLs, they demonstrated no apparent advantage exists of BLF IOLs over non-BLF IOLs in nAMD incidence.

As on the one hand it is possible to mitigate the phototoxic effect of blue light on the retina with the use of IOLs, it is at the same time possible to reduce the risk of retinal damage through the utilization of spectacle or contact lenses also in phakic eyes. In the last 10 years, there was a tremendous increase of compact fluorescent lamps and high-intensity light-emitting diodes (LEDs). Displays of both smartphones and tablets contain a large number of white-light LEDs, which appear white but have emissions at wavelengths corresponding to the peak of the blue light hazard function. This phenomenon may potentially lead to cumulative exposure to blue light and, consequently, to development of AMD. Standard spectacle lenses give protection against UV radiations and, with the addition of a yellow chromophore, it could be possible to decrease or eliminate blue light transmission. In order to ameliorate the retinal protection, antireflection interference coatings can be added to both anterior and posterior lens surfaces, attenuating a great part of the blue light [86]. A review by Lawrenson et al. [87] has investigated the relative benefits and potential harms of blue-light-blocking (BB) spectacle lenses for macular health. Currently, no studies support the hypothesis that BB spectacle lenses might protect against the onset of AMD.

In conclusion, there are conflicting results in the scientific literature regarding the use of blue-light-blocking spectacles or intraocular lenses to counteract the development of AMD. Although the retinotoxic effects of blue light and UV radiations are known, more studies are needed to demonstrate with certainty the benefits of using such devices to prevent the onset of AMD.

## 6. Conclusions

AMD is the leading cause of irreversible blindness in elderly people and represents nowadays a high social and economic burden for health systems. Intravitreal medications for nAMD and dietary supplementation to slow down the progression of the dry form are the mainstay of AMD treatment. Despite the good results obtained, the impact of AMD on worldwide population remains huge in terms of visual outcomes and quality of life. Implementing preventive strategies could be an alternative way while waiting for new therapies capable to improve the clinical course of AMD even more than we are able to do today. Modification of dietary habits with a balanced supply of antioxidants, such as vitamins C and E, lutein, and zeaxanthin, zinc, β- carotene, and omega-3 fatty acids has demonstrated to be able to reduce the risk of AMD. Specifically, the Mediterranean Diet has shown to be effective to prevent the onset of AMD. In addition, lifestyle modifications, firstly smoking cessation, but also doing regularly physical activity, reducing excessive body weight and avoiding obesity, and sleeping circa 7–8 house per night, have shown to be effective in decreasing the possibility to develop AMD. Lastly, the role of intraocular lenses and eyeglasses regarding the AMD prevention is still controversial and needs to be further elucidated, although the use of blue-light filters seems to protect the retina from photochemical damage.

In conclusion, a better lifestyle with balanced diet, no smoking, and regular physical activity might protect the eye from developing AMD. Therefore, it would be important for health systems to implement policies aimed to encourage a healthy lifestyle.

## Figures and Tables

**Table 1 jcm-10-03297-t001:** Daily nutritional supplementation based on the results of AREDS-2 study.

AREDS-2 Formulation	
Nutritional Supplement	Recommended Daily Dose
Vitamin E	400 IU
Vitamin C	500 mg
Lutein	10 mg
Zeaxanthin	2 mg
Copper	2 mg
Zinc	80 mg

**Table 2 jcm-10-03297-t002:** Nutrients and their food sources capable of preventing the onset of AMD.

Nutrients and Food Sources for AMD Prevention	
Nutrients	Food Sources
Vitamin C	Citrus fruits and juices, green peppers, broccoli, potatoes
Vitamin E	Whole grains, vegetable oil, eggs, nuts
β-Carotene	Carrots, kale, spinach
Zinc	Meat, poultry, fish, whole grains, dairy products
Lutein and Zeaxanthin	Chicken egg yolk, leafy green vegetables
Omega-3 (DHA and EPA)	Fish and seafood, nuts and seeds, plant oils

**Note**: Dietary modifications have demonstrated to be effective only in studies considering Western populations. Thus, these modifications do not necessary apply to people of other ethnicities.

**Table 3 jcm-10-03297-t003:** Mediterranean diet: a guide to daily food choices.

Mediterranean Diet	
Food Types	Frequency of Food Servings
Whole grains, bread, beans, legumes, nuts, and seeds	Daily (35% of total daily calories)
Vegetables and fruits	Daily (30% of total daily calories)
Extra virgin olive oil	Daily (10% of total daily calories)
Fish and seafoods Diary products, eggs, poultry, and yoghurt	Few times per week (20% of total daily calories)
Meat, sweets	Small amounts (5% of total daily calories)

**Note**: Dietary modifications have demonstrated to be effective only in studies considering Western populations. Thus, these modifications do not necessary apply to people of other ethnicities.

**Table 4 jcm-10-03297-t004:** Lifestyle modifications suggested to reduce the risk of AMD onset.

Lifestyle and AMD Prevention	
Activity/Condition	What to do to Prevent AMD Onset
Smoking	Cessation
Sleep duration	7–8 h per night
Weight	Reducing obesity and overweight to normal Body Mass Index BMI values (18.5–24.9); reducing waist circumferences
Physical activity	At least three hours of moderate- to low-intensity physical activity per week

## Data Availability

MDPI Research Data Policies at https://www.mdpi.com/ethics.

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
