# Peer review of "Prevention of the Onset of Age-Related Macular Degeneration"

_jcm, 2021, doi:10.3390/jcm10153297_

Round 1

Reviewer 1 Report

Carlo and Augustin describe the prevalence, incidence, impact on quality of life, social costs, and prevention of AMD. They summarize as follows. The prevalence of early and late AMD is higher in older people, and early AMD is more common in European white people than Asian population, whereas, the prevalence of late AMD are similar in all ethnic groups. AMD deteriorates quality of life by causing functional disability and other complications including falls, cardiovascular disease, depression, and cognitive impairment. AMD may increase the risk of mortality by 20 %. As the number of people with visual impairment and blindness due to AMD are increasing, socio-economic burden related to AMD is serious concern. For prevention of AMD, AERDS studies recommend nutritional supplementation therapy including vitamin C and E, lutein, zeaxanthin, copper, and zinc. The method to prevent the development of AMD includes taking healthy diet rich in antioxidants like Mediterranean diet, and avoiding unhealthful food (red and processed meats or savory and salty industrialized products), smoking, short or excessive sleep duration, obesity, and ultraviolet and blue light.

This is a well-written manuscript. Contents are rigorous and easy to follow.

Comments to be addressed.

  1. First of all, this review manuscript describes the prevalence and prevention of ‘drusen-driven’ AMD according to the previous reports. In other words, pachychoroid spectrum diseases are not considered. This should be mentioned in the Introduction.

  1. Section 2

The population-based studies and AREDS studies referenced in this manuscript were performed before the concept of pachychoroid spectrum diseases has been widely recognized. So these studies did not differentiate pachychoroid neovasculopathy from drusen-driven neovascular AMD. Pachychoroid spectrum diseases are much common in Asian than in western populations. Asians currently constitute 60% of the world’s population and likely will contribute most greatly to the global prevalence of AMD by 2040 [reference 2]. Before the introduction of the concept of pachychoroid spectrum diseases, pachychoroid neovasculopathy had often been diagnosed as AMD especially in Asian populations. This may partly explain why in the previous population-based studies, there was no difference in the prevalence of neovascular AMD in different ethnic groups even though the early AMD is more common in western than in Asian populations. Different genetic background is associated with different prevalence of pachychoroid spectrum disease as well as drusen-driven AMD.

The following paper may be a possible citation.

Yamashiro, K et al. Characteristics of pachychoroid diseases and age-related macular degeneration: multimodal imaging and genetic backgrounds. J Clin Med 9, 2034. https://doi:10.3390/jcm9072034 (2020).

  1. I think social costs should be briefly but more specifically described in the section 3.

  1. Dietary modifications described in the section 3 and 4 were based on the studies performed in the western populations. So readers should be noticed that these modifications in Table 2-4 do not necessarily apply to people in other ethnicity than western population.

Author Response

Dear Prof/Dr.,

Thank you very much for the important and kind suggestions. I provided to add in the manuscript the necessary corrections, as you suggested.

Attached is the new version of the manuscript

Reviewer 2 Report

The authors provide a comprehensive summary on prevention of onset of age-related macular degeneration (AMD). One major concern is that even the title and abstract focus only on the prevention of onset, there is a subtitle #4. Prevention of AMD progression: AREDS studies, in which authors spend almost one page to discuss the prevention of the disease progression. I would suggest the authors be clearer on the title, the abstract and anywhere necessary, that this review is about the prevention of onset and progression of AMD. Also, authors should polish their English writing more, since a few sentences are difficult to read or understand.

Specific comments are as below:

Page 1, line 25: should be ‘Recent studies predict that…’

Page 2, line 93: How can AMD lead to increase of diseases like cardiovascular diseases? Please make it clear or omit it.

Page 4, line 169: Would be easier to read if changed to ‘Both AREDS 1 and 2 studies have shown to prevent AMD progression by nutritional supplementation.’

Table 2, please add a column ‘proportion of daily consumption’, so that to give readers a better idea how much % of each kind of food should be consumed daily.

Author Response

Dear Prof./Dr.

Thank you very much for the important question. The article is focused on the prevention of AMD onset, as the title suggest. Furthermore, with regard to the section “AREDS studies” my intention was to underline the differences between previous studies, such as AREDS, which were focused on the prevention  of AMD progression with interventions during the course of the disease, and our study that is only focused on the prevention of AMD onset.

I have modified the title of the 4th section in order to better clarify the differences between our study (Prevention of AMD onset) and the other, above all the AREDS studies, where there were interventions (nutritional  supplementation) to stop the progression of the disease.

I hope I have made clear your legitimate doubts and I'm available for further modifications of the text.

Moreover, I modified the text with your kind suggestions.

Attached find the new version of the manuscript.

Best regards

Emiliano Di Carlo
